# Genetic Causal Association between Iron Status and Osteoarthritis: A Two-Sample Mendelian Randomization

**DOI:** 10.3390/nu14183683

**Published:** 2022-09-06

**Authors:** Jiawen Xu, Shaoyun Zhang, Ye Tian, Haibo Si, Yi Zeng, Yuangang Wu, Yuan Liu, Mingyang Li, Kaibo Sun, Limin Wu, Bin Shen

**Affiliations:** 1Department of Orthopedics, Orthopedic Research Institute, Sichuan University West China Hospital, 37# Guoxue Road, Chengdu 610041, China; 2Department of Orthopedics, The Third Hospital of Mianyang, Sichuan Mental Health Center, Mianyang 621000, China; 3Healthy Food Evaluation Research Center, West China School of Public Health and West China Fourth Hospital, Sichuan University, Chengdu 610041, China

**Keywords:** osteoarthritis, Mendelian randomization, causal association, genetic analyses

## Abstract

Objective: Observational studies have shown the association between iron status and osteoarthritis (OA). However, due to difficulties of determining sequential temporality, their causal association is still elusive. Based on the summary data of genome-wide association studies (GWASs) of a large-scale population, this study explored the genetic causal association between iron status and OA. Methods: First, we took a series of quality control steps to select eligible instrumental SNPs which were strongly associated with exposure. The genetic causal association between iron status and OA was analyzed using the two-sample Mendelian randomization (MR). Inverse-variance weighted (IVW), MR-Egger, weighted median, simple mode, and weighted mode methods were used for analysis. The results were mainly based on IVW (random effects), followed by sensitivity analysis. IVW and MR-Egger were used for heterogeneity testing. MR-Egger was also used for pleiotropy testing. Leave-one-SNP-out analysis was used to identify single nucleotide polymorphisms (SNPs) with potential impact. Maximum likelihood, penalized weighted median, and IVW (fixed effects) were performed to further validate the reliability of results. Results: IVW results showed that transferrin saturation had a positive causal association with knee osteoarthritis (KOA), hip osteoarthritis (HOA) and KOA or HOA (*p* < 0.05, OR > 1), and there was a negative causal association between transferrin and HOA and KOA or HOA (*p* < 0.05, OR < 1). The results of heterogeneity test showed that our IVW analysis results were basically free of heterogeneity (*p* > 0.05). The results of the pleiotropy test showed that there was no pleiotropy in our IVW analysis (*p* > 0.05). The analysis results of maximum likelihood, penalized weighted median and IVW (fixed effects) were consistent with our IVW results. No genetic causal association was found between serum iron and ferritin and OA. Conclusions: This study provides evidence of the causal association between iron status and OA, which provides novel insights to the genetic research of OA.

## 1. Introduction

Osteoarthritis (OA), which includes knee osteoarthritis (KOA) and hip osteoarthritis (HOA), is a highly prevalent chronic and degenerative joint disease that mainly occurs in the hip and knee. It’s featured by cartilage degeneration, changes in subchondral bone, and synovitis [1]. More than 500 million people worldwide are affected by OA, accounting for 7% of the world population [2]. Due to the aging of the population and the epidemic of obesity, the incidence of OA is on the rise [3]. In addition, the financial burden of OA patients is also increasing [4]. Unfortunately, there is no other effective treatment for OA except for the total joint replacement in the late stages [5]. OA is a complex disease caused by the interaction of multiple factors including trauma, metabolism, biological stress, and genetic susceptibility [6,7]. OA is polygenic and affects older individuals, with a recent genome-wide study of over 800,000 individuals adding 52 novel association signals [8]. A recent study found that OA shares a genetic predisposition to major depressive disorder [9]. However, the pathogenesis of OA has not been completely clarified. Therefore, it is of great significance to further explore the exact etiology of OA.

As an essential trace element of the human body, iron participates in oxygen transport, DNA replication, and adenosine triphosphate (ATP) synthesis, which plays a vital role in the growth and metabolism of all cells [10]. However, excess iron can lead to oxidative damage and dysfunction of cells and tissues (6). The body maintains iron homeostasis by regulating iron absorption, excretion, and storage [10]. Notably, iron homeostasis is also critical for joint health [11]. In recent years, the association between iron status and OA has attracted much attention, but the research on the causal association between iron status and OA is very limited. Some scholars point out that iron overload accelerates the development of OA [12,13]. On contrary, some believe that the iron level in the joints of OA patients may not be enough to cause corresponding pathological changes, because different concentrations of iron are affected by mechanical stimulation, inflammation, and oxidative stress [11]. Hence, further studies are warranted to demonstrate the causal association between OA and iron status, thereby conferring a theoretical basis for follow-up research and clinical treatment.

Mendelian randomization (MR) is an analysis of genetic variables that follows Mendel’s law of inheritance, which exploits single nucleotide polymorphisms (SNPs) as instrumental variables (IVs) to infer the causality of an observed association between a modifiable exposure and a clinically relevant outcome [14]. Alleles are randomly separated during meiosis, so MR can reduce the bias caused by confounding factors [15]. In addition, since genetic variation occurs before the disease and the order of the two cannot be reversed, therefore, MR can also avoid the interference of reverse causality [15]. Accumulating evidence has proven the reliability of MR. For instance, Yuan S et al. have demonstrated the genetic causal association between iron status and gout and rheumatoid arthritis (RA) through MR analysis [16]. In addition, Funck-Brentano T et al. have confirmed that there is a genetic causal association between body mass index (BMI) and knee and hip OA [17]. However, there is limited studies focus on the association between iron status and OA by using MR analysis.

This study used large-scale genome-wide association study (GWAS) data sets to analyze iron status indicators potentially causally related to OA through a two-sample MR study. This study may help to reveal the genetic characteristics and biological mechanisms of OA.

## 2. Materials and Methods

### 2.1. Data Sources

The summary-level data of the four indicators (serum iron, ferritin, transferrin saturation, and transferrin) of iron status came from the Genetics of Iron Status Consortium, and the data were of European descent (https://www.decode.com/summarydata, accessed on 1 June 2022). These data were obtained from 23,986 subjects of European descent in 11 cohorts of 9 participation centers. In addition, replication samples were obtained from 24,986 subjects of European descent in another 8 cohorts to confirm suggestibility and significant association. GWASs, genotype estimation, and quality control procedures (QCs) were performed in each cohort. In each cohort, QCs were performed on a single sample and SNP before the HAPMAP II (Release 22, NCBI Build36, dbSNP b126) or for InterAct, 1000 Genomes was entered. The association between genotype and input SNP and each iron phenotype was carried out through the additive model of alleles. A meta-analysis was undertaken to evaluate the SNP association and the *p*-value of gene analysis was obtained through the VEGAS (http://gump.qimr.edu.au/VEGAS/, accessed on 5 June 2022). The original GWASs had been approved by the relevant institutional review committee. A detailed description of the research process is shown in the published research [18].

The summary-level data of KOA, HOA, and K/HOA were from UK Biobank and Arthritis Research UK Osteoarthritis Genetics (arcOGEN) Consortium cohorts of European descent (http://www.arcogen.org.uk/, accessed on 7 June 2022). The summary-level data of KOA included 24,955 patients with KOA and 378,169 controls. The summary-level data of HOA included 15,704 patients with HOA and 378,169 controls. The summary-level data of K/HOA included 39,427 patients with K/HOA and 378,169 controls. A meta-analysis was conducted on the definition of OA using the aggregated statistics from the UK Biobank and arcOGEN cohorts, and the genome-wide significance was defined according to the combined *p*-value of the meta-analysis. A simple iterative program was used for physical aggregation to define independent signals in GWAS, and reciprocal approximate conditional analyses were performed. In order to better understand the sharing degree of genetic structure between OA and other complex traits, LD-score regression was performed in the LDHub pipeline (URLs). Multiple testing corrections were conducted using Benjamini Hochberg FDR and the effective number of independent traits. The original GWAS had been approved by the relevant institutional review committee. A detailed description of the research process is shown in the published research [19].

### 2.2. Instrumental Variable Selection 

To ensure effective IVs, the three basic model assumptions of MR analysis should be met. The selected IVs had a robust correlation with exposures. There were not any confounding factors for the IVs that met the exposure conditions. The selected IVs affected the outcomes only through exposures. First, we obtained SNPs strongly associated with exposure (*p* < 5 × 10^−8^). Then, we used the PhenoScanner database (http://www.phenoscaner.medschl.cam.ac.uk/, accessed on 12 June 2022) to manually screen and delete SNPs related to confounding factors and OA outcomes. When the SNP cannot be obtained from GWAS results, the proxy SNP was determined through the online platform LDlink (https://ldlink.nci.nih.gov/, accessed on 12 June 2022). In addition, palindrome SNPs were not included in our study. Finally, we identified three SNPs related to the serum iron (*p* < 5 × 10^−8^), four SNPs related to the ferritin (*p* < 5 × 10^−8^), four SNPs related to the transferrin saturation (*p* < 5 × 10^−8^), and eight SNPs related to the transferrin (*p* < 5 × 10^−8^). These SNPs acted as IVs to explore the potential causal association between exposures and outcomes. All these SNPs were not proxied, and palindrome SNPs (rs221834) among the four SNPs related to transferrin saturation were eliminated.

### 2.3. Mendelian Randomization Analysis

The “TwoSampleMR” R package (version 0.5.6, Stephen Burgess, Chicago, IL, USA) was used for two-sample MR analysis between exposures and outcomes. Inverse-variance weighted (IVW, random effects) were used as the main analysis methods. MR-Egger, weighted median, simple mode, and weighted mode were used as supplementary analysis methods. If the assumption that all included SNPs can be used as effective IVs is met, the IVW method provides an accurate estimate [16]. MR-Egger regression can detect and adjust the pleiotropy, but the estimation accuracy produced by this method is very low [20]. Weighted median gives an accurate estimate based on the assumption that at least 50% of IVs are valid [21]. Although simple mode is not as powerful as IVW, but it provides robustness for pleiotropy [22]. Weighted mode is sensitive to the difficult bandwidth selection for mode estimation [23].

### 2.4. Sensitivity Analysis

I^2^ index and Cochran’s Q statistic were adopted for IVW analysis, and the Rucker’s Q statistic were adopted for MR-Egger analysis to detect the heterogeneity of the effects of SNPs related to four iron status indicators on KOA, HOA, and K/HOA outcomes, and *p* > 0.05 indicates no heterogeneity [24]. MR-Egger regression is used to identify potential pleiotropy and evaluate the impact of pleiotropy on the risk estimation of the intercept test, and *p* > 0.05 indicates no pleiotropy [20]. Leave-one-SNP-out analysis is used to identify SNPs with potential impacts and evaluate the reliability of the results [25].

### 2.5. Further Validation of MR Results

To demonstrate the reliability of MR results, we used maximum likelihood, penalized weighted median, and IVW (fixed effects) for further analysis. Maximum likelihood is a traditional method with low standard error, which estimates the probability distribution parameters by maximizing the likelihood function [26]. Although bias may occur due to a limited sample size, the bias is so small that it can be biologically negligible [26]. IVW (fixed effects) is efficient when all IVs are valid and is sensitive to invalid IVs [27].

## 3. Results

### 3.1. Mendelian Randomization Analysis

IVW analysis showed that there was a positive causal association between transferrin saturation and KOA (*p* = 0.010, OR = 1.066) (Table 1, Figure 1). Weighted median analysis demonstrated that serum iron, ferritin, and transferrin saturation had a positive causal association with KOA (*p* < 0.05, OR > 1) (Table 1).

IVW analysis indicated a positive causal association between transferrin saturation and HOA (*p* = 0.042, OR = 1.155) and a negative causal association between transferrin and HOA (*p* = 0.018, OR = 0.915) (Table 1, Figure 2). MR-Egger analysis also showed a positive causal association between ferritin and HOA (*p* < 0.05, OR > 1) and a negative causal association between transferrin and HOA (*p* < 0.05, OR < 1). Weighted median results revealed that serum iron, ferritin, and transferrin saturation had a positive causality with HOA (*p* < 0.05, OR > 1), while transferrin had a negative causality with HOA (*p* < 0.05, OR < 1). Weighted mode results showed that ferritin and transferrin saturation had a positive causality with HOA (*p* < 0.05, OR > 1) (Table 1).

IVW analysis showed a positive causal association between transferrin saturation and K/HOA (*p* < 0.001, OR = 1.089), while a negative causal association between transferrin and K/HOA (*p* = 0.028, OR = 0.953) (Table 1, Figure 3). MR-Egger analysis indicated that there was a negative causal association between transferrin and K/HOA (*p* < 0.05, OR < 1). Weighted median results revealed that serum iron, ferritin, and transferrin saturation had a positive causality with K/HOA (*p* < 0.05, OR > 1). Weighted mode analysis demonstrated a positive causality between ferritin and K/HOA (*p* < 0.05, OR > 1), but a negative causality between transferrin and K/HOA (*p* < 0.05, OR < 1) (Table 1).

In summary, our results were mainly based on IVW analysis: transferrin saturation had a positive causal association with KOA, HOA, and K/HOA, and transferrin had a negative causal association with HOA and K/HOA. 

### 3.2. Sensitivity Analysis

Sensitivity analysis was conducted to verify the reliability of IVW results. IVW and MR-Egger test for heterogeneity showed that there was no heterogeneity in MR analysis results between transferrin saturation and KOA and K/HOA (*p* > 0.05), and there was heterogeneity in MR analysis results between transferrin saturation and HOA (*p* < 0.05) (Table 2). The results of the IVW test for heterogeneity showed that there was heterogeneity in MR analysis results between transferrin and HOA and K/HOA (*p* < 0.05) (Table 2).

The MR-Egger test for heterogeneity showed that there was heterogeneity in MR analysis results between transferrin and HOA (*p* < 0.05), and there was no heterogeneity in MR analysis results between transferrin and K/HOA (*p* > 0.05) (Table 2). MR-Egger regression results showed that there was no pleiotropy in MR analysis results (*p* > 0.05) (Table 2). The results of the leave-one-SNP-out analysis indicated that rs1800562, rs8177272, and rs17376530 might have a potential impact on IVW results (Appendix A).

### 3.3. Further Validation of MR Results

Since the leave-one-SNP-out analysis found SNPs that might have a potential impact on IVW analysis results, so we further verified the IVW results. The results of maximum likelihood, penalized weighted median, and IVW (fixed effects) revealed that transferrin saturation had a positive causal association with KOA (*p* < 0.05, OR > 1) and HOA (*p* < 0.05, OR > 1) (Figure 4). The results of Maximum likelihood and IVW (fixed effects) demonstrated a negative causal association between transferrin and HOA (*p* < 0.05, OR < 1) (Figure 4). Maximum likelihood, penalized weighted median, and IVW (fixed effects) results showed a positive causality between transferrin saturation and K/HOA (*p* < 0.05, OR > 1), and Maximum likelihood and IVW (fixed effects) results indicated a negative causality between transferrin and K/HOA (*p* < 0.05, OR < 1) (Figure 4).

## 4. Discussion

MR analysis demonstrated that transferrin saturation had a positive causal association with KOA, HOA, and K/HOA, while transferrin had a negative causal association with HOA and K/HOA but had no correlation with KOA. No genetic causal association was found between serum iron and ferritin and KOA, HOA, or K/HOA. MR has great potential for analyzing the causal associations between diseases and traits. This study is the first to investigate the genetic causal associations between OA and iron status, making a significant contribution to the study of the mechanism of OA.

At present, accumulating studies have documented the association between transferrin saturation and hereditary hemochromatosis (HH), but relatively little is known about the role of transferrin saturation in OA [28]. HH, an autosomal recessive disorder caused by the mutation of the homeostatic iron regulator (HFE) gene responsible for regulating iron homeostasis, is mainly manifested as a systemic iron overload [28,29]. Most HH patients present joint symptoms similar to OA [11]. A longitudinal cohort study of HH patients has shown that persistent high transferrin saturation is positively correlated with the severity of joint destruction [30]. In comparison to patients that are compound heterozygous HH (C282Y/H63D), patients with homozygote HH (C282Y/C282Y) patients have a higher transferrin saturation level and a higher risk of OA [31]. Intriguingly, a prospective cross-sectional study has also suggested that there is no significant difference in transferrin saturation between patients with severe HOA and normal subjects [32]. However, the majority view supports the positive correlation between transferrin saturation and the risk of OA, which is consistent with our research results.

In an animal study, the exposure of primary chondrocytes from HFE-deficient mice to a high-iron environment not only increases the content of matrix metalloproteinases related to pathological changes of OA, but also up-regulates the expression of transferrin [33]. Mitochondrial dysfunction and oxidative stress are critical for iron overload-induced cartilage degeneration [34]. Free iron is toxic, and the main function of transferrin is to bind free iron in circulation [35]. When the transferrin level is decreased or transferrin binding capacity is overloaded, excess free iron generates hydroxyl radicals through the Fenton reaction, which damages mitochondria, and produces reactive oxygen species (ROS), eventually resulting in cell damage and death [34,35]. In addition, ROS can react on mitochondria, leading to mitochondrial dysfunction and oxidative stress [34]. Notably, mitochondrial dysfunction and oxidative stress contribute to the development of OA [36,37]. It can be seen that the low transferrin level can promote the progress of OA by causing mitochondrial dysfunction and oxidative stress. The transferrin level is negatively correlated with OA severity, which is consistent with our findings that transferrin has a negative genetic causal association with OA.

In this study, we found that ferritin had no genetic causal association with OA. What’s more, a recent MR study explored the causal associations between serum nutritional factors and OA revealed that ferritin has no genetic correlation with KOA and HOA [38], which is consistent with our results. However, it has been reported that the ferritin level is positively correlated with KOA, and the imaging severity of KOA is enhanced with the increase of ferritin level [39]. The serum ferritin level is also positively correlated with the degree of cartilage damage in KOA [40]. The results of this study showed that there was no causal association between ferritin and OA from the perspective of genetics, but it did not rule out the possibility that other factors played a potential regulatory role in the correlation between ferritin and OA. For example, ferritin status is known to be well affected by inflammation, which plays a positive role in the progression of cartilage injury and the pathogenesis of OA [41]. The increased iron storage reflected by the higher ferritin concentration can produce reactive nitrogen and toxic oxygen free radicals, which have become catabolic factors in the process of OA [42]. In addition, the concentration of ferritin promotes inflammation and tissue damage through neutrophil infiltration, with deleterious effects on synovial tissue in OA patients [43,44]. 

Iron status and the occurrence of inflammatory illness are closely related. Due to the complicated associations between iron status and inflammatory illness such as OA, the causal associations between iron status and OA are still elusive. For example, when iron status is disrupted, iron undergoes sequential redox activities and is capable of releasing numerous free radicals, including superoxide anion, hydroxyl radical, and hydrogen peroxide [45]. This oxidative stress-induced injury and subsequent inflammatory response are central to inflammatory illness [46]. However, the inflammatory environment also plays an important role in regulating iron status through the hepcidin–ferroporin axis [47]. To sum up, it is hard to say which is exposure and which is outcome in previous studies. In this study, by using a large sample size of European descent, we used MR analysis to explore the causal association between iron status and OA and found transferrin saturation and transferrin had causal associations with OA. This is the first time to systematically explore the causal association between iron status and OA. 

This study also has some limitations. First of all, the subjects in this study are all of the European descent, so it should be prudent to apply the results of this study to other races. The confounding factors, such as age, gender, and other environmental confounding factors also have a certain impact on MR analysis. In addition, due to the different detection kits of these four indicators (serum iron, ferritin, transferrin saturation, and transferrin), there are existing some systemic errors which cannot be avoided. To avoid errors, further studies should use detection kits from the same company. At last, this studies only explore the causal associations between iron status (exposure) and OA (outcome). The reverse causal associations between OA and iron status will be discussed in the further studies.

Even so, this study still puts forward novel insights into the genetic causal association between iron homeostasis and OA from the perspective of genetics, thereby providing guidance for future research.

## 5. Conclusions

To our knowledge, the current study is the largest genetic association study between iron status-related indicators (serum iron, ferritin, transferrin saturation, and transferrin) and OA (KOA, HOA, and K/HOA). It is found that transferrin saturation has a positive causal association with OA, and transferrin has a negative causal association with OA. The results of this study can provide novel insights to the genetic research of OA.

## Figures and Tables

**Figure 1 nutrients-14-03683-f001:**
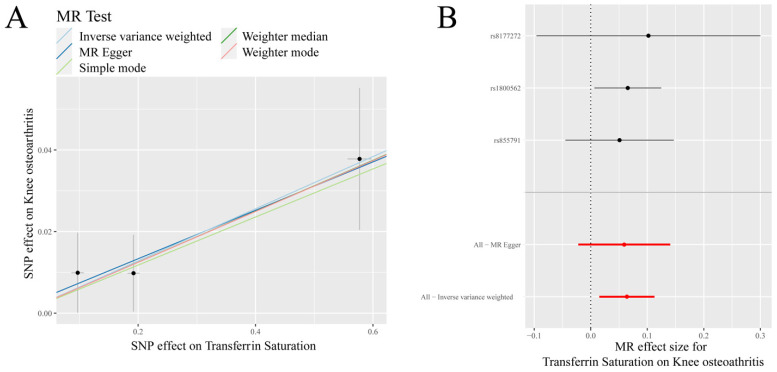
MR results of transferrin saturation and knee osteoarthritis (KOA): (**A**) scatter plot of genetic correlations of transferrin saturation and KOA using different MR methods. The slopes of line represent the causal effect of each method, respectivel; (**B**) forest plot of the causal effects of transferrin saturation associated SNPs on KOA. The red and black dot/bar indicate the causal estimate of transferrin saturation level on risk of patients with KOA.

**Figure 2 nutrients-14-03683-f002:**
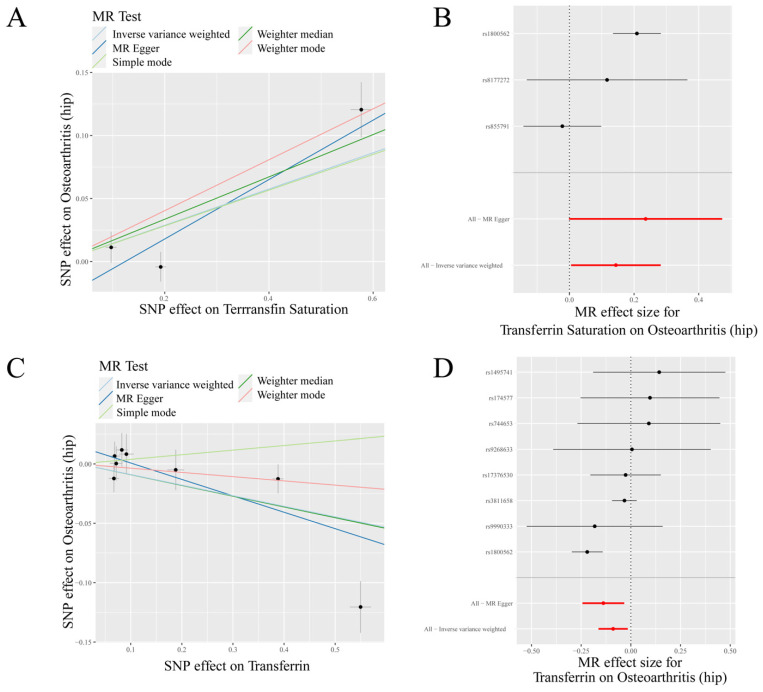
MR results of transferrin saturation, transferrin and hip osteoarthritis (HOA): (**A**) scatter plot of genetic correlations of transferrin saturation and HOA using different MR methods. The slopes of line represent the causal effect of each method, respectively; (**B**) forest plot of the causal effects of transferrin saturation associated SNPs on HOA; (**C**) scatter plot of genetic correlations of transferrin and HOA using different MR methods. The slopes of line represent the causal effect of each method, respectively; and (**D**) forest plot of the causal effects of transferrin associated SNPs on HOA. (**B**): The red and black dot/bar indicate the causal estimate of transferrin saturation level on risk of patients with HOA. (**D**): The red and black dot/bar indicate the causal estimate of transferrin level on risk of patients with HOA.

**Figure 3 nutrients-14-03683-f003:**
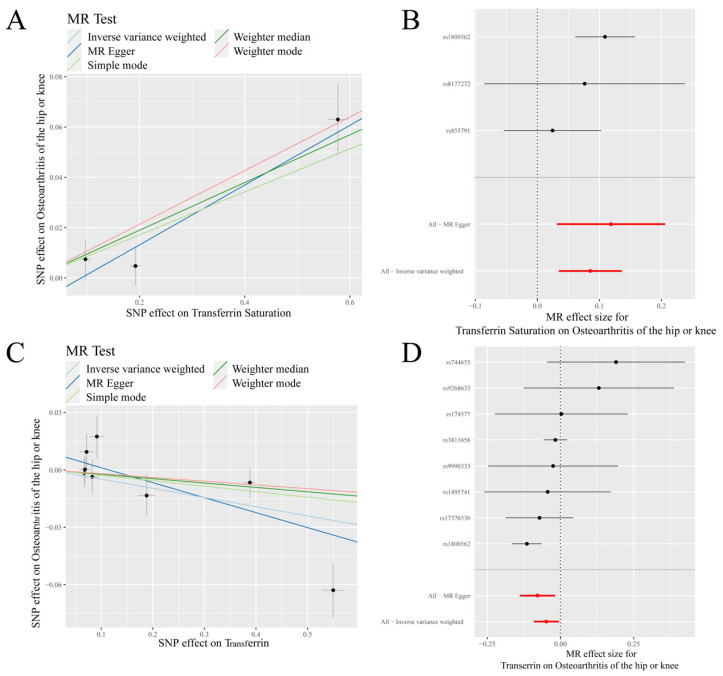
MR results of transferrin saturation, transferrin and knee/hip osteoarthritis (K/HOA): (**A**) scatter plot of genetic correlations of transferrin saturation and K/HOA using different MR methods. The slopes of line represent the causal effect of each method, respectively; (**B**) forest plot of the causal effects of transferrin saturation associated SNPs on K/HOA; (**C**) scatter plot of genetic correlations of transferrin and K/HOA using different MR methods. The slopes of line represent the causal effect of each method, respectively; and (**D**) forrest plot of the causal effects of transferrin associated SNPs on K/HOA. (**B**): The red and black dot/bar indicate the causal estimate of transferrin saturation level on risk of patients with K/HOA. (**D**): The red and black dot/bar indicate the causal estimate of transferrin level on risk of patients with K/HOA.

**Figure 4 nutrients-14-03683-f004:**
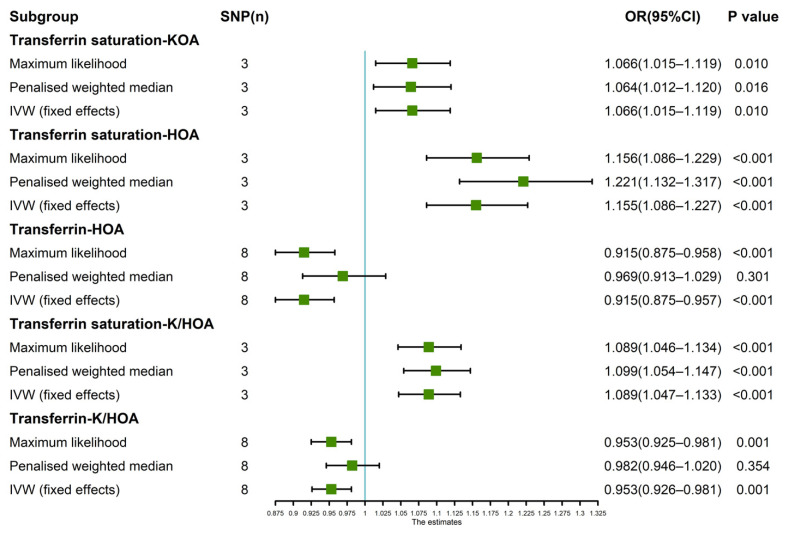
Estimated causal effects between iron status indicator and OA using different MR methods.

**Table 1 nutrients-14-03683-t001:** Mendelian randomization (MR) analysis of osteoarthritis (OA) and iron status indicators.

	Method	Serum Iron			Ferritin			Transferrin Saturation			Transferrin		
		SNP (n)	OR (95%CI)	*p* Value	SNP (n)	OR (95%CI)	*p* Value	SNP (n)	OR (95%CI)	*p* Value	SNP (n)	OR (95%CI)	*p* Value
**KOA**	**MR Egger**	3	1.176 (1.035–1.337)	0.244	4	1.194 (0.782–1.822)	0.498	3	1.061 (0.978–1.151)	0.389	8	0.942 (0.878–1.010)	0.146
	**Weighted median**	3	1.078 (1.008–1.153)	0.028	4	1.185 (1.020–1.375)	0.026	3	1.064 (1.013–1.119)	0.014	8	0.959 (0.920–1.000)	0.051
	**IVW**	3	1.068 (0.986–1.156)	0.106	4	1.054 (0.860–1.292)	0.610	3	1.066 (1.015–1.119)	0.010	8	0.964 (0.920–1.011)	0.128
	**Simple mode**	3	1.080 (0.995–1.172)	0.207	4	1.163 (0.798–1.695)	0.491	3	1.061 (0.992–1.135)	0.228	8	0.918 (0.851–0.991)	0.065
	**Weighted mode**	3	1.082 (1.003–1.168)	0.178	4	1.186 (1.007–1.398)	0.134	3	1.064 (1.009–1.123)	0.149	8	0.958 (0.922–0.995)	0.063
**HOA**	**MR Egger**	3	1.482 (0.970–2.264)	0.320	4	2.122 (1.589–2.835)	0.036	3	1.266 (0.999–1.604)	0.302	8	0.871 (0.784–0.967)	0.042
	**Weighted median**	3	1.142 (1.004–1.299)	0.043	4	1.265 (1.004–1.593)	0.047	3	1.183 (1.095–1.277)	<0.001	8	0.913 (0.852–0.979)	0.010
	**IVW**	3	1.155 (0.896–1.489)	0.266	4	1.358 (0.966–1.910)	0.079	3	1.155 (1.005–1.326)	0.042	8	0.915 (0.850–0.985)	0.018
	**Simple mode**	3	0.920 (0.772–1.097)	0.452	4	0.997 (0.718–1.384)	0.986	3	1.152 (0.905–1.467)	0.369	8	1.040 (0.893–1.211)	0.633
	**Weighted mode**	3	0.983 (0.732–1.319)	0.917	4	1.767 (1.377–2.267)	0.021	3	1.224 (1.130–1.325)	0.038	8	0.965 (0.896–1.039)	0.377
**K/HOA**	**MR Egger**	3	1.246 (1.080–1.438)	0.204	4	1.434 (1.020–2.014)	0.174	3	1.126 (1.032–1.229)	0.229	8	0.925 (0.871–0.982)	0.043
	**Weighted median**	3	1.095 (1.028–1.167)	0.005	4	1.164 (1.002–1.353)	0.047	3	1.099 (1.051–1.150)	<0.001	8	0.977 (0.935–1.022)	0.316
	**IVW**	3	1.094 (0.973–1.231)	0.134	4	1.146 (0.922–1.424)	0.220	3	1.089 (1.035–1.146)	<0.001	8	0.953 (0.913–0.995)	0.028
	**Simple mode**	3	1.007 (0.878–1.155)	0.932	4	0.917 (0.660–1.274)	0.641	3	1.089 (0.996–1.192)	0.203	8	0.972 (0.895–1.056)	0.523
	**Weighted mode**	3	1.106 (1.025–1.193)	0.139	4	1.310 (1.138–1.509)	0.033	3	1.113 (1.058–1.170)	0.054	8	0.981 (0.937–1.027)	0.429

Note: KOA knee osteoarthritis, HOA hip osteoarthritis, SNP single nucleotide polymorphism, MR mendelian randomization, IVW inverse variance weighting, OR odds ratio, CI confidence interval.

**Table 2 nutrients-14-03683-t002:** Sensitivity analysis of the Mendelian randomization (MR) analysis results of osteoarthritis (OA) and iron status indicator.

Exposure		Transferrin Saturation	Transferrin Saturation	Transferrin	Transferrin Saturation	Transferrin
Outcome		KOA	HOA	HOA	K/HOA	K/HOA
**IVW (heterogeneity)**	***p* value**	0.898	0.006	0.009	0.196	0.035
	**Q**	0.214	10.279	18.860	3.255	15.098
**MR Egger (heterogeneity)**	***p* value**	0.659	0.020	0.020	0.188	0.071
	**Q**	0.195	5.402	15.001	1.735	11.637
**MR Egger (pleiotropy)**	***p* value**	0.912	0.516	0.260	0.521	0.230
	**intercept**	0.001	−0.029	0.015	−0.011	0.009

Note: KOA knee osteoarthritis, HOA hip osteoarthritis, SD standard deviation, MR mendelian randomization, SNP single nucleotide polymorphism, MR-PRESSO MR-Pleiotropy Residual Sum and Outlier method, IVW inverse variance weighting.

## Data Availability

The summary-level data of iron status came from the Genetics of Iron Status Consortium (https://www.decode.com/summarydata, accessed on 12 June 2022). The summary-level data of OA were from UK Biobank and Arthritis Research UK Osteoarthritis Genetics (arcOGEN, http://www.arcogen.org.uk/, accessed on 12 June 2022). The MR analysis was performed through the TwoSampleMR packages (version 0.5.6).

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
