# Peer review of "Genetic Causal Association between Iron Status and Osteoarthritis: A Two-Sample Mendelian Randomization"

_nutrients, 2022, doi:10.3390/nu14183683_

Round 1

Reviewer 1 Report

Authors studied genetic association between iron status and OA (KOA, HOA, and K/HOA). Transferrin saturation (TSAT) has a positive causal association with OA, and transferrin has a negative causal association with OA. Looking back some previous reports about joint disease and iron status, this result can be acceptable concept. However, there are some concerns as follows.

Major comment

Based on the iron metabolism in the human body, iron status is well affected by inflammation. Thus, a research related with iron status must elucidate the high inflammation condition or explain the status of inflammation. However, there is no description about inflammation. I strongly suggest to mention about inflammation status. WBC or CRP are major blood markers of inflammation. Moreover, Authors described that ferritin had no genetic association with OA. As authors know, ferritin status is known to be well affected by inflammation. In this regard, inflammation markers may give authors a hint about unexpected ferritin result. Ferritin status in chronic rheumatoid arthritis is known to reveal the wide range. Additionally, there is another considering point about ferritin. The detection kits of ferritin are known to differ by companies, which sometimes comes to a discussion for researches of iron clinically. Thus, authors should consider the possibility about difference of ferritin kits.

Minor comment

There is no abbreviation of KOA and HOA. Authors should add an explanation.

Characters of each figure is too small. I recommend to enlarge them.

Author Response

Reviewer 1

Authors studied genetic association between iron status and OA (KOA, HOA, and K/HOA). Transferrin saturation (TSAT) has a positive causal association with OA, and transferrin has a negative causal association with OA. Looking back some previous reports about joint disease and iron status, this result can be acceptable concept. However, there are some concerns as follows.

Major comment

  1. Based on the iron metabolism in the human body, iron status is well affected by inflammation. Thus, a research related with iron status must elucidate the high inflammation condition or explain the status of inflammation. However, there is no description about inflammation. I strongly suggest to mention about inflammation status. WBC or CRP are major blood markers of inflammation.

Response: Thank you for the helpful comments.

            We are very sorry for the confused places. Iron status and the occurrence of inflammatory illness are closely related. Due to the complicated associations between iron status and inflammatory illness such as OA, the causal associations between iron status and OA are still elusive. For example, when iron status is disrupted, iron undergoes sequential redox activities and is capable of releasing numerous free radicals, including superoxide anion, hydroxyl radical, and hydrogen peroxide [1]. This oxidative stress-induced injury and subsequent in-flammatory response are central to inflammatory illness [2]. However, the inflammatory environment also plays an important role in regulating iron status through the hepcidin-ferroporin axis [3]. To sum up, it is hard to say which is exposure and which is outcome in previous studies. In this study, by using a large sample size of European descent, we used MR analysis to explore the causal association between iron status and OA and found transferrin saturation and transferrin had causal associations with OA. This is the first time to systematically explore the causal association between iron status and OA.

            Unfortunately, there is no study identified the genetic associations between iron status and inflammatory markers (WBC and CRP). Therefore, we cannot demonstrate the genetic associations between iron status and specific inflammatory markers clearly, their associations should be identified in our further studies by using MR analysis.

            Per your guidance, we added the association between iron status and inflammation status in the “Discussion” section. Please see in page 10, line 294-306 in the revised manuscript.

  1. Moreover, Authors described that ferritin had no genetic association with OA. As authors know, ferritin status is known to be well affected by inflammation. In this regard, inflammation markers may give authors a hint about unexpected ferritin result. Ferritin status in chronic rheumatoid arthritis is known to reveal the wide range.

Response: Thank you for the helpful comments.

            We are very sorry for the confused places. Studying the genetic causal association between ferritin and OA is of great significance to the research on the etiology, mechanism and treatment of OA. This is the first study to explore the causality between ferritin and OA through MR analysis. No evidence of positive or negative causality between ferritin and OA was found, indicating that ferritin had no causality with OA in genetics. However, it did not rule out the possibility that other factors played a potential regulatory role in the correlation between ferritin and OA. For example, ferritin status is known to be well affected by inflammation, which plays a positive role in the progression of cartilage injury and the pathogenesis of OA [4]. The increased iron storage reflected by the higher ferritin concentration can produce reactive nitrogen and toxic oxygen free radicals, which have become catabolic factors in the process of OA [5]. In addition, the concentration of ferritin promotes inflammation and tissue damage through neutrophil infiltration, with deleterious effects on synovial tissue in OA patients [6-8]. Per your guidance, we added the associations among ferritin, inflammation, and OA in the “Discussion” section. Please see in page 10, line 287-293 in the revised manuscript.

  1. Additionally, there is another considering point about ferritin. The detection kits of ferritin are known to differ by companies, which sometimes comes to a discussion for researches of iron clinically. Thus, authors should consider the possibility about difference of ferritin kits.

Response: Thank you for the helpful comments.

            We are very sorry for the confused places. Due to the different detection kits of these four indicators (serum iron, ferritin, transferrin saturation, and transferrin), we cannot eliminate this systemic errors. Per your guidance, we added this as a limitation of our study in the “Discussion” section. Please see in page 10, line 307-316 in the revised manuscript.

Minor comment

  1. There is no abbreviation of KOA and HOA. Authors should add an explanation.

Response: Thank you for the helpful comments.

            We are very sorry for the confused places. Per your guidance, we added the explanation of KOA (knee osteoarthritis) and HOA (hip osteoarthritis). Please see in page 1, line 28,40 in the revised manuscript.

  1. Characters of each figure is too small. I recommend to enlarge them.

Response: Thank you for the helpful comments.

We are very sorry for the confused places. Per your guidance, we enlarged the characters of each figure. Please see in Figure 1-4 in the revised manuscript.

References

  1. Elsayed, ME.; Sharif, MU, Stack, AG. Transferrin Saturation: A Body Iron Biomarker. Adv Clin Chem. 2016, 75,71-97.
  2. Lugrin, J.; Rosenblatt-Velin, N.; Parapanov, R, Liaudet, L. The role of oxidative stress during inflammatory processes. Biol Chem. 2014, 395,203-230.
  3. Ueda, N, Takasawa, K. Impact of Inflammation on Ferritin, Hepcidin and the Management of Iron Deficiency Anemia in Chronic Kidney Disease. Nutrients. 2018, 10.
  4. Nugzar, O.; Zandman-Goddard, G.; Oz, H.; Lakstein, D.; Feldbrin, Z, Shargorodsky, M. The role of ferritin and adiponectin as predictors of cartilage damage assessed by arthroscopy in patients with symptomatic knee osteoarthritis. Best Pract Res Clin Rheumatol. 2018, 32,662-668.
  5. de Boer, TN.; van Spil, WE.; Huisman, AM.; Polak, AA.; Bijlsma, JW.; Lafeber, FP, Mastbergen, SC. Serum adipokines in osteoarthritis; comparison with controls and relationship with local parameters of synovial inflammation and cartilage damage. Osteoarthritis Cartilage. 2012, 20,846-853.
  6. Carroll, GJ.; Sharma, G.; Upadhyay, A, Jazayeri, JA. Ferritin concentrations in synovial fluid are higher in osteoarthritis patients with HFE gene mutations (C282Y or H63D). Scand J Rheumatol. 2010, 39,413-420.
  7. Brissot, P.; Ropert, M.; Le Lan, C, Loréal, O. Non-transferrin bound iron: a key role in iron overload and iron toxicity. Biochim Biophys Acta. 2012, 1820,403-410.
  8. Heiland, GR.; Aigner, E.; Dallos, T.; Sahinbegovic, E.; Krenn, V.; Thaler, C.; Weiss, G.; Distler, JH.; Datz, C.; Schett, G, Zwerina, J. Synovial immunopathology in haemochromatosis arthropathy. Ann Rheum Dis. 2010, 69,1214-1219.

Reviewer 2 Report

The manuscript is interesting, and is related to the investigation of the causal genetic factors assoc. with OA and iron status. Please check the tables (?). Besides, the supplementary document does not have a legend. There are several corrections (see the attached document) marked in the pdf document at rows: 30, 70, 80, 95, 126, 235.

Author Response

The manuscript is interesting, and is related to the investigation of the causal genetic factors assoc. with OA and iron status. Please check the tables (?). Besides, the supplementary document does not have a legend. There are several corrections (see the attached document) marked in the pdf document at rows: 30, 70, 80, 95, 126, 235.

Response: Thank you for the helpful comments.

We are very sorry for the confused places. First, we noticed that the peer-review version of our manuscript did not contains Table1 (page 5) and Table2 (page 8). Per your guidance, we added the Table1 and Table2 in the revised manuscript. Please see the revised manuscript. In addition, we added the figure legends to illustrate the supplementary figure (page 15). Please see supplementary figure in the revised manuscript. Per your guidance, we have corrected the mistakes you mentioned in the pdf document. Please see in line 40, 81, 92, 101, 135, 256 in the revised manuscript.

Round 2

Reviewer 1 Report

Although there are some limitations, my concern is diminished. Thank you for your response.